# An Innovative EEG-Based Pain Identification and Quantification: A Pilot Study

**DOI:** 10.3390/s24123873

**Published:** 2024-06-14

**Authors:** Colince Meli Segning, Rubens A. da Silva, Suzy Ngomo

**Affiliations:** 1Department of Applied Sciences, UQAC (Université du Québec à Chicoutimi), Chicoutimi, QC G7H 2B1, Canada; colince.meli-segning1@uqac.ca; 2Biomechanical and Neurophysiological Research Laboratory in Neuro-Musculoskeletal Rehabilitation (Lab BioNR), Department of Health Sciences, UQAC (Université du Québec à Chicoutimi), Chicoutimi, QC G7H 2B1, Canada; rdsilva@uqac.ca; 3Centre Intégré de Santé et Services Sociaux du Saguenay-Lac-Saint-Jean (CIUSSS SLSJ), Specialized Geriatrics Rehabilitation Services at the La Baie Hospital, CIUSSS-SLSJ, Saguenay, QC G7H 7K9, Canada

**Keywords:** quantification of pain, EEG signal, pain identification and quantification indicator (Piq), beta EEG frequency band, electrodes on contralateral motor regions

## Abstract

Objective: The present pilot study aimed to propose an innovative scale-independent measure based on electroencephalographic (EEG) signals for the identification and quantification of the magnitude of chronic pain. Methods: EEG data were collected from three groups of participants at rest: seven healthy participants with pain, 15 healthy participants submitted to thermal pain, and 66 participants living with chronic pain. Every 30 s, the pain intensity score felt by the participant was also recorded. Electrodes positioned in the contralateral motor region were of interest. After EEG preprocessing, a complex analytical signal was obtained using Hilbert transform, and the upper envelope of the EEG signal was extracted. The average coefficient of variation of the upper envelope of the signal was then calculated for the beta (13–30 Hz) band and proposed as a new EEG-based indicator, namely Piq_β_, to identify and quantify pain. Main results: The main results are as follows: (1) A Piq_β_ threshold at 10%, that is, Piq_β_ ≥ 10%, indicates the presence of pain, and (2) the higher the Piq_β_ (%), the higher the extent of pain. Conclusions: This finding indicates that Piq_β_ can objectively identify and quantify pain in a population living with chronic pain. This new EEG-based indicator can be used for objective pain assessment based on the neurophysiological body response to pain. Significance: Objective pain assessment is a valuable decision-making aid and an important contribution to pain management and monitoring.

## 1. Introduction

Background: Chronic pain continues to be a global public health issue, with an estimated prevalence of approximately 30% in adults worldwide [1,2,3]. Among other issues related to this important global health problem, assessing chronic pain remains a challenge. Traditionally, pain assessment relies on hetero- or self-reported pain [4,5,6]. Although verbal description is only one of several behaviors to express pain, these possibilities for assessing chronic pain unfortunately lose all their robustness when it comes to people who have difficulty expressing themselves, such as the older, those with cognitive disorders, or those suffering from a psychological disorder [7,8]. This is why our research over the last few years has aimed towards a proposal for pain assessment, which is not based on the verbal or behavioral expression of pain. It is consensual that the experience of pain is related to alterations in brain excitability [9,10]. Specifically, motor regions seem to be largely involved, even though the underlying neurological mechanisms still need to be clarified [11,12]. Therefore, our proposal echoes the community’s understanding of the mechanisms involved in the production and maintenance of pain. The relationship between beta band activity and pain perception is supported by previous research indicating that beta oscillations are involved in sensory and motor processing, which are crucial in the experience of pain [13,14,15]. Beta band activity is often found to be altered in conditions of pain, reflecting changes in cortical excitability. For example, Teixeira et al. [16] recently evaluated beta (β) oscillations as a potential objective marker for pain assessment. Twelve adult right-handed males with chronic neuropathic pain and 10 matched controls participated in this pilot study. Participants underwent pain assessment using a visual analog scale. The authors then calculated the global power spectrum within the low beta frequency sub-band (13–20 Hz) and the high beta frequency sub-band (20–30 Hz). Their results showed that the global power spectrum was significantly lower in patients compared to controls. Additionally, the visual analog scale for pain was negatively correlated with the global power spectrum in both the low beta (R = −0.931, *p* = 0.007) and high beta bands (R = −0.805; *p* = 0.053). We recently (2022) published a proof-of-concept paper for pain identification, based on brain signal activity, collected via electroencephalography (EEG) [17], with a small homogeneous sample of participants living with moderate chronic pain (*n* = 4). EEG signal collection was conducted in two parts: (1) at rest and (2) in the active state, that is, participants executing a visuo-motor task, and three frequency bands (alpha (8–13 Hz), beta (13–30 Hz), and gamma (30–43 Hz)) were analyzed. The results showed that beta band measurements at rest offered better results in the pain condition tested, that is, in moderate pain status. Currently, our research team is interested in better generalizing these data to different pain contexts: tonic pain induced by capsaicin cream and induced by thermal pain and in a real chronic pain condition. This demonstration contributes to testing our algorithm’s sensitivity to deliver meaningful results in real-life chronic pain conditions in real patients. The literature, along with our own observations, indicates that the beta band is the most appropriate for studying pain identification and quantification. Consequently, the present work has been conducted with data collected in this frequency band.

Objective: The goal of this pilot study was to propose an innovative scale-independent measure for pain identification and quantification based on an analysis of the envelope of the EEG signal [18,19,20]. Specifically, the objectives achieved were: (1) developing a methodological approach for identifying and quantifying pain in real-world settings; (2) quantifying a threshold to detect the presence of pain; (3) analyzing the relationship between the new EEG-based indicator and self-reported pain, assessed using a verbal numerical rating scale (VNRS); and (4) examining the effect size of medication on brain activity within this new framework for chronic pain assessment. Based on a previous study [17], we hypothesized that the pain message on the EEG signal would amplify the envelope variability of the EEG signal and that dynamic changes in envelope variation in the presence of pain could be proportional to the magnitude of pain experienced.

## 2. Materials and Methods

### 2.1. Participants

Pain was quantified in three groups of volunteers by convenience (see Table 1): Group 1 related to seven healthy participants (six men, one woman: 24–45 years of age) who submitted in two main experimental conditions: “No pain” and “With tonic pain” induced by a capsaicin cream. Group 2 included 15 healthy participants (14 men, one woman: 22–45 years of age) who experienced experimentally induced thermal pain, and Group 3 included 66 individuals (21 men, 45 women: 15–78 years of age) living with chronic pain, such as shoulder pain, fibromyalgia, and low back pain. For this group, in addition to sociodemographic data, daily medication use was also ascertained. Among the 66 participants with real chronic pain, 36 took medications known to be centrally acting [21]. Therefore, a sub-objective was to verify the effect size of medication on brain activity using the new approach. The local Research Ethics Committee (CER #2023-1200 and CER # 2023-1307) approved the study, and the participants provided written consent for their participation.

### 2.2. Experimental Procedures

Given that pain mechanisms evolve through brain activity, the main physiological material used in our study was EEG signals. For EEG collection, the participants were seated at rest, with eyes opened focused on a black dot located 1.5 m away on the wall in front of them at eye level. An EMOTIV wireless electroencephalographic headset was placed on the participant’s head to collect brain activity; electrodes were positioned according to the 10–20 international system. Pain intensity was assessed using a 0–10 verbal numerical rating scale (VNRS), where 0 corresponded to no pain and 10 to the worst imaginable pain [22].

#### 2.2.1. Procedure for Group 1: Healthy Participants— “No Pain” and “with Pain”

In the condition “With pain”, pain was induced with a topical application of a ~1 cm wide band of capsaicin cream (1%) (~1 mm thick) on the right upper trapezius in the same previous group. The capsaicin cream contained 1% capsaicin, the hot ingredient in chili peppers. This cream induced localized moderate tonic pain at the site of application. The aim of this study was to create experimental pain similar to musculoskeletal pain. This cream was used in accordance with previous studies [23,24]. The plateau of pain was attained approximately 30 min after application and corresponded to the moment when the participant consecutively showed the same pain intensity three times (pain intensity was assessed every 2 min).

Data from this group were collected in two conditions: (1) “No pain and (2) “With pain”. EEG signals were collected first in the “No pain” condition and second in the “With pain” condition. Analyses were performed on the EEG signals collected during the last 60 s for each condition. Given that the pain plateau was not reached until 30 min after the application of the capsaicin cream, we chose the last 60 s of the recordings from each of the two conditions to obtain two comparable moments.

#### 2.2.2. Procedure for Group 2: Healthy Participants Subjected to Thermal Experimental Pain

A thermal stimulus was applied to the non-dominant forearm using a thermode (Figure 1) initially heated to 44 °C. Knowing that, on contact with the skin, the temperature of the thermode would begin to decrease, we held the thermode on the skin for 5 min to allow it to reach the baseline temperature at around 32 °C [25,26,27]. An initial temperature of 44 °C was chosen based on previous work, where it is commonly accepted that at this temperature most subjects report a transformation of heat sensation into pain [28,29]. Three variables (pain stimulus (°C), pain score/10, and EEG signals) were collected simultaneously for 5 min (Figure 2).

#### 2.2.3. Procedure for Group 3: Participants Living with Chronic Pain

Data were collected in a clinical setting. EEG recordings lasted 5 min (300 s) and every 30 s (metronome Bip), and the score of pain intensity felt by the participant was recorded, as well as in group 2.

### 2.3. EEG Data Acquisition

A wireless EEG device, an Emotiv EPOC X 16-channel headset (Emotiv Systems Inc., San Francisco, CA, USA), including 14 active and two ground electrodes, was used. Impedance was maintained in a 10–20 KΩ range by properly wetting the sponge electrodes with a saline solution and controlling the electrode contact quality map, which should be green during EEG data collection to ensure the good quality of the EEG signal. EEG data were acquired with an internal sampling frequency of 2048 Hz. Data were then digitalized using an embedded 16-bit analog-to-digital converter and down-sampled to 128 samples per second before being transmitted to the acquisition computer. The digitalized EEG signals were online-filtered by the EPOC X hardware with a 5th-order digital sinc filter using a bandpass of 0.2–45 Hz and a notch digital filter at 60 Hz (for North America) to eliminate power line frequency noise.

#### 2.3.1. EEG Data Preprocessing

The preprocessing of the EEG signals was performed in three steps, as illustrated in Figure 3 and Figure 4, and is described in the following subsections: Step 1—EEG signal filtering in two substeps: (a) Direct Current (DC) offset removal, also known as baseline correction, and (b) EEG artifact removal; Step 2—EEG frequency band selection; and Step 3—Normalization of filtering EEG signals in two substeps (min–max and baseline normalization). Signals from electrodes positioned over the bilateral motor regions (FC6/T8 on the right side and FC5/T7 on the left side) were of interest. These electrodes cover the recommended positioning on the motor regions according to the EPOC X headset [30]. These electrodes are located over the frontal and temporal regions of the scalp, corresponding to areas involved in motor function. Moreover, previous studies have reported the involvement of motor cortex areas in the pain process [31,32,33]. The preprocessing was conducted using the MATLAB software, version: 9.10.0.1602886 (R2021a) (MathWorks Inc., Natick, MA, USA).

##### EEG Signals Filtering

The first step of pre-processing was to remove noise from the EEG signals. The direct current (DC) voltage offset, that is, the offset of a signal from 0, was first removed using the simplest method consisting of subtracting the average value (approximately 4200 µV) from the entire selected data channel. The second substep of filtering was to remove the remaining artifacts due to eye blinks or eye movements, and the electromyography (EMG) signal and all well-known noise, such as poor electrode contact quality, were identified during the experiments. To this end, an outlier detection and replacement filter was used by applying Matlab’s “filloutliers” function [34,35]. First, the outlier values defined as EEG values that were over 1.5 interquartile ranges above the upper quartile (75%) or below the lower quartile (25%) were detected using the quartiles find method. Second, the linear interpolation of neighboring non-outlier values method was used to replace the detected outlier values.

##### EEG Frequency Band Selection

The second step of preprocessing was the selection of the beta (13–30 Hz) EEG frequency sub-band over the electrode positioned contralateral to the pain site for further analyses [36,37,38,39,40]. Indeed, previous studies and our previous work have reported the relevance of beta for pain identification [17]. Finally, a 5th-order IIR Butterworth bandpass filter was used for the selected frequency band [41].

##### EEG Signal Normalization

The third preprocessing step was EEG signal normalization in two substeps. The first substep was to adjust for the inter-variability of EEG data within the selected frequency band by scaling the EEG data using min–max normalization. This substep involves placing artifact-free EEG signals in the interval between 0 and 1 [42,43]. Min–max normalization performs a linear transformation of the original data values while preserving the relationships between them. The equation to achieve this processing is as follows [44]:(1)$${\mathrm{EEG}}_{\min-\max}^{\beta }=\frac{{\mathrm{EEG}}_{\mathrm{AF}}^{\beta }-\min\left({\mathrm{EEG}}_{\mathrm{AF}}^{\beta }\right)}{\max\left({\mathrm{EEG}}_{\mathrm{AF}}^{\beta }\right)-\min\left({\mathrm{EEG}}_{\mathrm{AF}}^{\beta }\right)}$$
where ${\mathrm{EEG}}_{\mathrm{AF}}^{\beta }$ and ${\mathrm{EEG}}_{\min-\max}^{\beta }$ correspond to the artifact-free EEG signal and min–max normalized signal, respectively, and $\min\left({\mathrm{EEG}}_{\mathrm{AF}}^{\beta }\right)$ and $\max\left({\mathrm{EEG}}_{\mathrm{AF}}^{\beta }\right)$ denote the minimum and maximum of all artifact-free EEG signals within the beta frequency band. Finally, the second substep was baseline normalization, which re-scales the previous min–max normalization values by the weight of each single min–max normalized EEG value. It divides each min–max normalized EEG value using a selected reference. The resulting equation is as follows:(2)$${\mathrm{EEG}}_{N}^{\beta }=\frac{{\mathrm{EEG}}_{\min-\max}^{\beta }}{{\mathrm{Ref}}^{\beta }}$$
where ${\mathrm{EEG}}_{N}^{\beta }$ represents the normalized EEG signals for the beta frequency band and ${\mathrm{Ref}}^{\beta }$ is the reference, which is defined as the mean of the min–max EEG signals in the reference interval for the beta frequency band. In this study, ${\mathrm{Ref}}^{\beta }$ was considered as the first 60 s of EEG collection for each participant.

#### 2.3.2. Extracting Pain-Related Feature from EEG Signals to Pain Identification and Quantification Indicator (Piq)

As shown in Figure 3 and Figure 4, after the EEG signals were filtered and normalized, the complex analytical signal $z_{\beta }\left(t\right)$, which considers non-stationarity and nonlinearity as these are the characteristics of the EEG signal, was obtained using the Hilbert transform (Figure 4) [19,45]. This type of analytic signal is a two-dimensional signal whose value at some instant in time is specified by two parts: a real part (Figure 4a) and an imaginary one (Figure 4b) [46]. The upper envelope of the EEG signal was then extracted as the absolute of the analytical digital signal (Figure 4c) [19]. Finally, the coefficient of variation of the upper envelope (CVUE) was calculated for the beta frequency band (13–30 Hz). The coefficient of variation represents the ratio of the standard deviation to the mean and is a useful statistic for comparing the degree of variation from one data series to another. The Piq indicator was computed in beta (Piq_β_) in five steps.

(1)The first step was the estimation of the analytic EEG signals for the beta frequency band ($z_{\beta }\left(t\right)$) using Hilbert transform, as follows:(3)$$z_{\beta }\left(t\right)=s_{\beta }\left(t\right)+{j\tilde{s}}_{\beta }\left(t\right)$$
where ${\tilde{s}}_{\beta }\left(t\right)=H\left(s_{\beta }\left(t\right)\right)$ represents Hilbert transform, the steps of which are illustrated in Figure 4.

(2)The second step was the extraction of the upper envelope (UE) of the EEG signals for the beta frequency band (${\mathrm{UE}}_{\beta }$), defined as the absolute value of the analytic signal, as follows:(4)$${\mathrm{UE}}_{\beta }\left(t\right)=\left|z_{\beta }\left(t\right)\right|=\sqrt{s_{\beta }^{2}\left(t\right)+{j\tilde{s}}_{\beta }^{2}\left(t\right)}$$

The upper envelope (UE) of a given cortical oscillation reflects the energy range over time [45]. The UE was high when energy was high. Because of the time-variant behavior and nonlinear neuronal system responsible for generating the EEG signals, preprocessed EEG signals were segmented in sliding windows (epochs) of 1 s duration, and ${\mathrm{UE}}_{\beta }$ was then calculated within each epoch.

(3)The third step was the estimation of the coefficient of variation of the upper envelope in the beta EEG frequency band (CVUE_β_). To this end, the mean and standard deviation (std) of ${\mathrm{UE}}_{\beta }$ were computed in each epoch to obtain CVUE_β_ as follows:(5)$${\mathrm{CVUE}}_{\beta }\left(\%\right)=\frac{\mathrm{std}\left({\mathrm{UE}}_{\beta }\left(t\right)\right)}{\mathrm{mean}\left({\mathrm{UE}}_{\beta }\left(t\right)\right)}\times 100$$

Low CVUE values reflect more stable sinusoidal oscillations, that is, more neuronal synchronization or inhibition [47]. In contrast, a high CVUE corresponds to neuronal desynchronization [18,47], that is, less inhibition (more facilitation).

(4)The fourth step consisted of smoothing ${\mathrm{CVUE}}_{\beta }$ using a 15th-order Savitzky–Golay filter.

In each sliding window, the so-called edge effects can be preserved and affect the CVUE values [48,49]. Smoothing filters, such as a Savitzky–Golay filter, make it possible to correct inter alia spikes present in the data [50,51]. The Savitzky–Golay filter is a least-squares smoothing filter (digital polynomial filter). Its working principle involves replacing each value with a new value, previously obtained from polynomial fitting, which is performed with basic linear least-squares fitting to the 2*k* + 1 neighboring points, where the value *k* could be equal to or greater than the order of the applied polynomial. The more neighbors that are applied, the smoother the final signal [50]. It smooths the fluctuations in data and increases the signal-to-noise ratio (SNR) without significantly distorting the analyzed data [51,52]. In this study, a 15th-order Savitzky–Golay filter was used.

(5)The fifth step was the calculation of the pain identification and quantification (Piq) indicator in the beta frequency band (Piq_β_).

The mean smoothing ${\mathrm{CVUE}}_{\beta }$ was calculated and proposed as a pain identification and quantification (Piq) indicator. The higher the Piq (%), the higher the magnitude of pain.

### 2.4. Statistical Analysis

Objective 1 was already completed in Section 2.3.2 (Figure 3 and Figure 4), showing each step of the methodological approach proposed to extract the relevant indicator for the identification and quantification of pain. The mean and standard deviation values were used for descriptive statistics in relation to the second objective (2), that is, to determine the threshold quantified for the identification of the presence of pain. The correlation coefficient was used to determine the relationship between the main variables, that is, Piq_β_ × pain scores, across groups to address objective (3), that is, the relationship between the proposed approach and self-reported pain on a verbal numerical rating scale. Finally, to achieve the secondary objective, we utilized effect size (ES) and clinical difference (Δ) to measure the impact of medication on brain activity, thus fulfilling this sub-objective, that is, the magnitude of the effect of medication acting on brain activity in the new approach. ES was calculated based on the Cohen criteria, i.e., *d* = 0.2 to 0.49 is small, *d* = 0.5 to 0.79 is medium, and *d* ≥ 0.8 is large [53]. A decrease in pain of less than 15% was judged as a non-important change, of 15% or more as a minimally important change, of 30% or more as a moderately important change, and of 50% or more as a substantially important change [54]. The significance level of the tests was set at *p* < 0.05, and all statistical analyses were conducted using SPSS version 24 (IBM Corp., Armonk, NY, USA).

## 3. Results

### 3.1. New Approach to Identify and Quantify Pain

In this study, as illustrated in Figure 3 and Figure 4, a method for extracting pain-related features from EEG signals, that is, a pain identification and quantification indicator (Piq), was proposed. This innovative methodological approach inspired by the variation in the morphology of EEG signals in animals [47] and humans [20], which utilizes beta (β) brain rhythm, is proposed for pain identification and quantification. Piq_β_ is proposed as a pain indicator for pain identification and quantification in the beta band frequency. As a reminder, the higher the Piq_β_ (%), the higher the magnitude of pain.

### 3.2. Descriptive Statistics of Groups to Meet Objective (2) i.e., Determine the Threshold Quantified for the Identification of the Presence of Pain

A visual inspection of the results showed that Piq_β_ ≥ 10% was indicative of the presence of pain (Table 2 and Table 3). Therefore, Piq ≤ 10% corresponded to little or no pain.

#### 3.2.1. Group 1: Healthy Participants—“No Pain” and “with Pain”

Table 2 shows the descriptive statistics of Group 1 subjected to capsaicin application.

In the pain condition, participants had moderate pain on average (pain score ≥ 4 on the VNRS). All participants in the “no pain” condition had a Piq_β_ of less than 10% and, conversely, in the experimental pain condition.

#### 3.2.2. Group 2: Healthy Participants Submitted to Thermal Pain

The protocol for Group 2 was specifically designed to evaluate the effectiveness of the proposed algorithm in tracking pain changes in very short timeframes, achieving Objective 1 (the identification and quantification of pain). A visual inspection of the graph (Figure 5) showed that Piq_β_ can track the magnitude of pain. In addition, this study provides evidence that 5 min of EEG signal collection is sufficient to address the variation in pain magnitude.

For Group 2, we performed normalization by bringing all the values of each pain indicator between 0 and 1 while maintaining the distances between the values and allowing for the comparison of the three variables on the same scale (Figure 5). This min–max normalization was performed using Equation (6), as follows:(6)$$\;Normalized\;value=\frac{Z-minimum\left(Z\right)}{maximum\left(Z\right)-minimum\left(Z\right)}\;$$
where z represents the value of each pain indicator (score intensity, level of pain stimulus, and Piq_β_).

This study showed a simultaneous decrease in the level of pain stimulus, pain score intensity, and Piq_β_ indicator.

#### 3.2.3. Group 3: Participants Living with Chronic Pain

This third study also confirmed that the pain threshold of the EEG-based indicator Piq_β_ is 10% to identify individuals with chronic pain as well as sensitive to quantify the extent of pain (Piq_β_ higher than 10%, Table 3 and Figure 6).

### 3.3. Results for Objective (3) i.e., the Relationship between the Proposed Approach for the Identification and Quantification of Pain (Piq_β_) and Self-Reported Pain (Score/10)

For healthy participants with capsaicin pain (group 1), no correlation test was performed because of the small sample size. In Group 3, for the participants living with chronic pain, a significant and strong positive correlation (*r* = 0.69, *p* < 0.0001) was found between Piq_β_ and pain scores from the VNRS, while a moderate relationship between variances (R^2^ = 0.47) was observed.

Considering all three studies, our findings indicate that Piq_β_ (%) appears to be affected by pain variability. This implies that varying levels of pain could potentially affect Piq_β_ measurements or that Piq_β_ might serve as a sensitive gauge for changes in pain across the individuals under study. Nevertheless, it is crucial to emphasize that this observation warrants further investigation to ascertain a causal relationship.

### 3.4. Results for the Secondary Objective, i.e., the Effect Size of Medication Acting on Brain Activity on the New Approach

The participants in group 3 were divided into two subgroups according to their medication: (1) with (*n* = 36) and (2) without centrally acting (*n* = 30). The effect sizes and clinical differences were then calculated (Table 4).

According to Cohen’s classification, the effect size of medication on the pain score was moderate (*d* = 0.61). The subgroup of participants taking centrally acting medication achieved a significant reduction in pain scores (*p* = 0.016), and this reduction (55.8%) was considered to be a substantially important change [55]. In addition, in the same group, the effect size of the medication on Piq_β_ was moderate (*d* = 0.51). The Piq_β_ indicator displayed significantly lower values (*p* = 0.041), with an average decrease of 18.5%, suggesting a minimally important change. Overall, these results showed that medication had a significant negative and moderate effect on the magnitude of pain. Consistently, we noted that Piq_β_ (%) and the pain score (/10) were higher in the absence of medication acting on brain activity.

## 4. Discussion

This pilot study aimed to propose (1) developing a methodological approach for identifying and quantifying pain in real-world settings; (2) quantifying a threshold to detect the presence of pain; (3) analyzing the relationship between the new EEG-based indicator and self-reported pain, assessed using a verbal numerical rating scale (VNRS); and (4) examining the effect size of medication on brain activity within this new framework for chronic pain assessment.

### 4.1. New Methodological Approach for Pain Identification and Quantification

Our methodological approach demonstrates the promising potential for objective pain identification and quantification based on the analysis of the envelope of EEG signals, which is related to relevant aspects of EEG signal morphology and remains more stable over time and conditions [17,20,47]. The main advantage of this methodology is that it keeps the EEG signal complex, thus preserving almost all of its intrinsic properties, such as non-stationarity and nonlinearity, offering the maximum amount of relevant information. This new approach, which is based on the morphological stability parameters of the EEG signal, provides great methodological robustness. In addition, EEG systems are typically multichannel in nature, resulting in numerous features being extracted from multichannel signals [56]. It is widely recognized that employing a large number of channels or features in pain identification or quantification poses practical limitations in real-life scenarios [57,58,59]. Such limitations include prolonged experimental setup times, subject discomfort, and increased computational complexity associated with handling multichannel EEG recordings [60]. Given these constraints, there is considerable value in developing a portable pain identification and quantification method that focuses on the optimal feature from the representative channel. Thus, the complexity and dynamic nature of pain identification and quantification can be circumvented by optimizing the hardware setup. This approach not only streamlines the experimental process but also reduces the burden on the subjects and computational resources. Therefore, feature selection is vital for EEG-based pain identification and quantification. For this purpose, we applied a simple and easy-to-implement algorithm to an EEG signal collected for five minutes over the motor cortex region, without altering patient comfort in the clinical setting. Although the proposed indicator seems optimal for pain identification and quantification, it is important to validate it using different machine learning classifiers to propose an automatic system that produces satisfactory accuracy.

### 4.2. Identification of Pain

We propose a quantitative indicator that we called “Piq” for Pain identification and quantification. Although there is no gold standard for quantitative measures of pain, the self-reported intensity score of pain remains one of the relevant comparative variables in the context of our study. Our results (group 1 et 3) suggest that a Piq greater than or equal to 10% (Piq ≥ 10%) is indicative of the presence of pain. Therefore, for Piq < 10%, there was little to no pain. In our recent study that utilized the current method, we observed that among healthy participants not experiencing pain, the Piq value was consistently below 10% [17].

### 4.3. Quantification of Pain

We propose Piq as an indicator to reflect the magnitude of pain experienced by an individual. Our results showed a strong positive correlation between Piq_β_ and self-reported pain in both experimental and real chronic pain conditions. These results suggest the ability of Piq_β_ to track the presence and extent of acute, subacute, and chronic pain. In other words, the higher the self-reported pain score, the higher the Piq_β_ (%), and vice versa. Of course, this study comprised experimental acute pain, given the practical improbability of collecting acute pain data in a real-world setting. Our results are similar to those of Nir et al. [38], who evaluated the perception of tonic pain using a thermal contact-heat simulator, showing that increased peak alpha frequency values derived from EEG recordings of the resting state and noxious conditions were correlated with higher pain intensity. However, to the best of our knowledge, no study has proposed the use of all EEG signals to detect pain. Given that pain is a complex biopsychosocial experience, it became evident that the appropriate method for conducting the analyses was to use EEG signals with complex values for the selected sub-band rather than EEG signals with real values. Complex-valued signals seemed appropriate, as they captured the complex information of the pain process, integrating the multidimensional aspects of this phenomenon. In the chronic pain group (*n* = 66), our results showed a moderate relationship between the variances of self-reported pain scores and Piq_β_ (%) (R^2^ = 0.47; *p* < 0.001). Note that R^2^, the coefficient of determination, measures the percentage of variation in the dependent variable that is explained by a variation in the independent variable [61]. However, there is no rule for interpreting the strength of R^2^ in terms of clinical relevance; a low R^2^ can still provide a useful clinical model [62,63].

Our hypothesis about the moderate variance found in participants with chronic pain is that Piq_β_ (%) reflects a combination of cerebral mechanisms implemented in the context of a painful condition, whereas the self-reported pain score is an expression of the overall biopsychosocial experience of pain. Several studies have shown alterations in cerebral excitability or even a cerebral reorganization in the context of prolonged pain [64,65]. Our results showed a positive relationship between Piq_β_ and the duration of pain (*p* < 0.05, *r* = 0.36), as well as between the duration of pain and pain score (*p* < 0.05, *r* = 0.25) in the chronic pain group (*n* = 66). However, these relationships are weak, reinforcing the idea of alteration in cortical excitability rather than cerebral reorganization. In general, studies have shown that pain is accompanied by a decrease in cortical inhibition [18,47,66,67,68,69]; therefore, we hypothesized that an increase in Piq_β_ is indicative of less cortical inhibition. One possible approach to validate this hypothesis is to integrate EEG with TMS to enable the measurement of cortical inhibition.

### 4.4. Effect Size of Brain-Acting Medication on Pain Measurement (Piq_β_ (%))

As we identified the presence and quantification of pain magnitude from brain activity, centrally acting medication may influence the outcome. We found that the effect size of medication was moderate on the pain score (*d* = 0.61) as well as on Piq_β_, suggesting that Piq_β_ (*d* = 0.51) reflects the effectiveness of pain treatments in this sample, ranging from moderate to low [55]. Persistent pain in a sample of 66 participants was typically treated with medications such as antidepressants, anticonvulsants, and opioids [70]. The clinical difference in the Piq_β_ indicator between both subgroups was 18.5% i.e., a minimally important change in favor of the centrally acting medication in the subgroup.

Taken together, our results show that the Piq_β_ indicator can track pain variation via brain activity, even in the presence of a centrally acting treatment, which can be advantageous for monitoring pain treatment in various contexts, from outpatient to postoperative orthopedic care.

### 4.5. Perspective

Several questions require additional research to fill these gaps. For example, what is the minimum Piq_β_ value (%) that indicates a clinically relevant change in pain magnitude? Alternatively, does the measurement time frame with the proposed approach change depending on the type of treatment, or, for example, what is the optimal time for the identification and quantification of pain after a physical, pharmacological, or surgical intervention? For minimum clinical change, it is recommended that two or more different methods be used to evaluate the clinical importance of improvement or worsening for chronic pain clinical trial outcome measures [54,71]. Pain assessment is generally classified into 11 levels on the NRS [72]: 0–1: Weak or no pain. 1–6: Mild to moderate pain, invisible. 7–10: For intense to severe pain, several physiological signs make it possible to identify that a person is experiencing pain. This study showed that the presence of pain was set at 10% for Piq_β_. Thus, work needs to be carried out to establish Piq_β_ values that can be considered as the minimally relevant clinical change in Piq_β_ between the two interventions. In the forthcoming phases of this project, we intend to expand the participant sample to include more individuals experiencing pain, and we will assess their pain as part of a protocol aiming to test the reliability of the Piq_β_ indicator. This endeavor holds significance, as enhancing clinical relevance will offer a more profound insight into its practical applicability. Furthermore, we aimed to confirm the observed Piq_β_ threshold value of 10% across the three studies through a static analysis conducted in a study with a broader sample size.

### 4.6. Limits

The main limitation of the present pilot study is the absence of real pain data showing large variations in the levels of pain, as for experimental thermal pain. This would have enabled a more robust corroboration that Piq_β_ tracks acute pain. Nonetheless, it is important to bear in mind that this study serves as a pilot study aimed at testing our algorithm designed to identify and quantify chronic pain under real-life conditions. The algorithm outlined in our 2022 paper, incorporating a sample comprising four individuals experiencing fibromyalgia pain [17], is being implemented and evaluated in this context. Our primary focus was the objective assessment of chronic pain, ranging from mild to moderate intensity, which may be accompanied by episodes of intense pain. It is an invisible disease that is difficult to objectively assess. In line with the 11^th^ International Classification of Diseases, our work aims to advance the recognition of chronic pain as a health problem on its own [73].

## 5. Conclusions

In conclusion, we have successfully met the objectives of this pilot study, which included: (a) developing a new method for identifying and quantifying pain, (b) establishing a threshold to detect the presence of pain, (c) exploring the correlation between the new EEG-based indicator and self-reported pain using a verbal numerical rating scale (VNRS), and (d) assessing the impact of medication on brain activity as a secondary objective within the new chronic pain assessment. Specifically, from a methodological perspective, here are the key parameters from the pilot study that are essential for the algorithm’s functionality: (1) Five (5) min for EEG collection at rest posture provides sufficient material for the purpose of analysis, without necessarily affecting the sufferer’s comfort; (2) one (1) electrode positioned above the motor regions contralateral to the site of pain is sufficient and allows us to respect the physiological decussation of the ascending and descending pathways of information; (3) the frequency band of interest of EEG signal is beta (13–30 Hz), and this frequency is the one that allows us to optimally capture useful and interesting information in the painful state; (4) Piq_β_ threshold at 10%, i.e., Piq_β_ ≥ 10% is indicative of the presence of pain; and (5) the higher the Piq_β_ (%), the higher the extent of the pain.

This finding could have strong implications for the population living with pain, specifically persistent pain, and we are thinking here of all the workers on prolonged sick leave for musculoskeletal pain, as well as of healthcare professionals, insurance companies, the pharmaceutical industry, etc. An objective assessment is a valuable decision-making aid and an important contribution to pain management and monitoring.

## Figures and Tables

**Figure 1 sensors-24-03873-f001:**
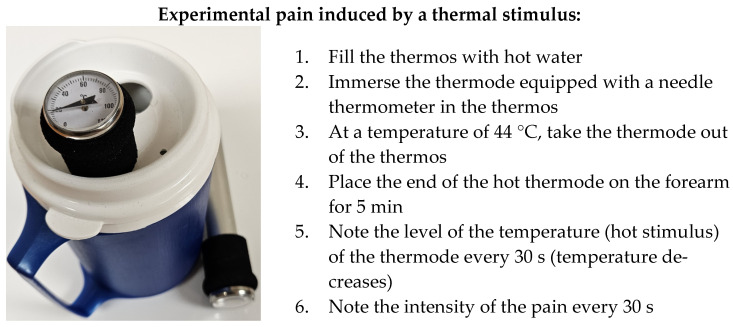
The thermal stimulus kit.

**Figure 2 sensors-24-03873-f002:**
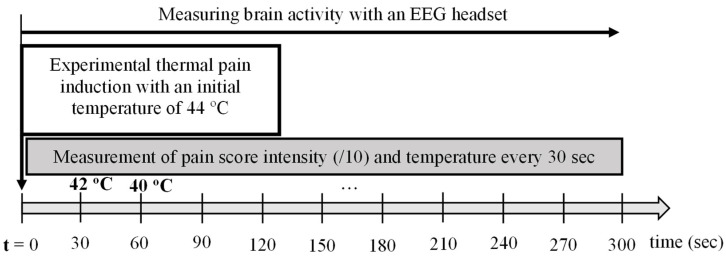
Experimental design—Thermal stimulus, pain score, and EEG recording during 300 s or 5 min.

**Figure 3 sensors-24-03873-f003:**
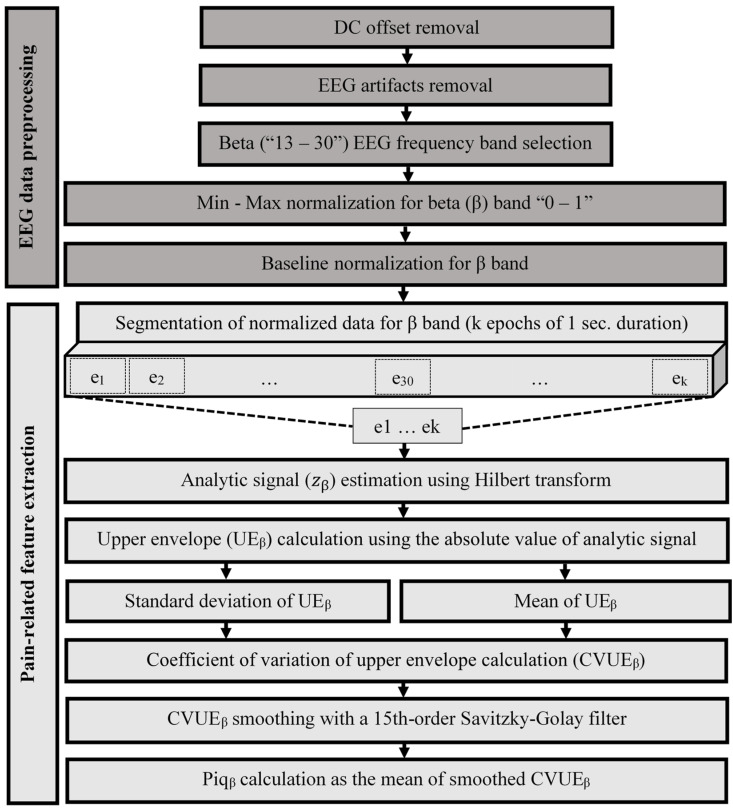
Methodological approach from the filtering of the EEG signal, through the estimation of the coefficient of variation of the upper envelope in beta (CVUE_β_), to the calculation of pain identification and quantification (Piq_β_).

**Figure 4 sensors-24-03873-f004:**
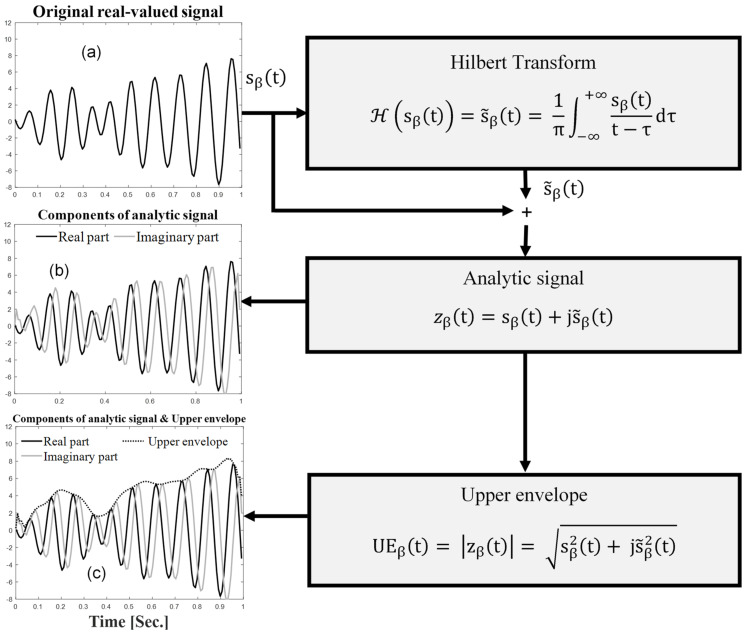
Methodological steps showing the detail of the application of the Hilbert transform until the extraction of the upper envelope. (**a**)—original real-valued signal, (**b**)—real and imaginary parts of analytic signal, (**c**)—superposition of real and imaginary parts of analytic signal and upper envelope of original signal.

**Figure 5 sensors-24-03873-f005:**
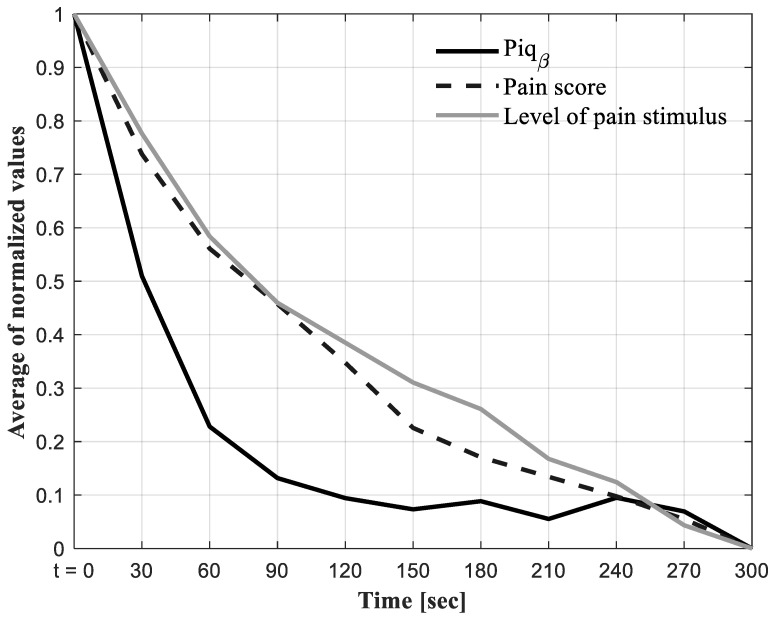
Normalized mean [0–1] for all participants (*n* = 15) of the three variables: (1) Normalized pain score intensity (black dotted line curve), (2) normalized level of pain stimulus (grid curve), and (3) normalized pain identification and quantification in beta frequency band (Piq_β_) (black curve).

**Figure 6 sensors-24-03873-f006:**
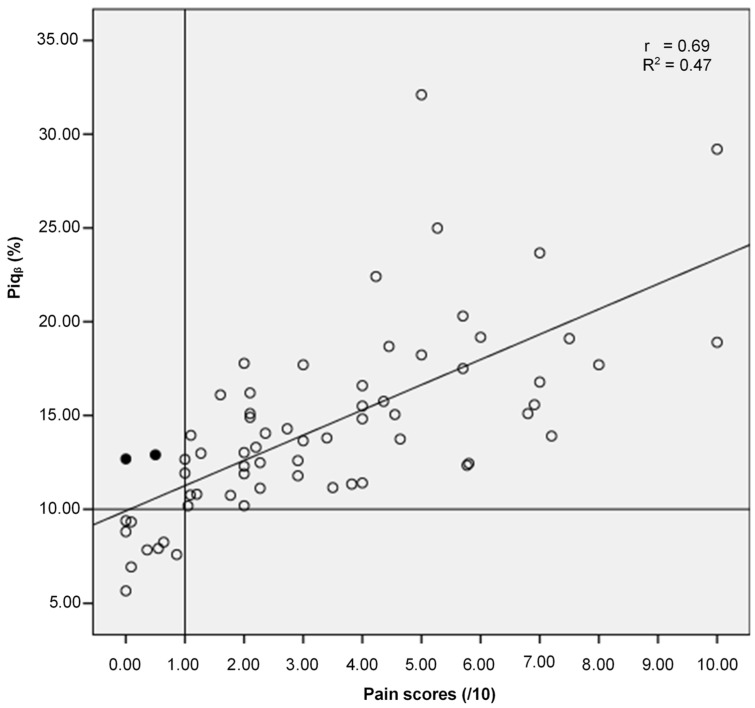
Scatter plot—Piq_β_ indicator and pain score. 100% of participants living with chronic pain show a Piq_β_ ≥ 10%. The two solid points represent participants who reported a pain score lower than 1/10 but had a Piq_β_ indicator ≥10%. The hollow points represent participants whose pain scores are consistent with their Piq_β_ indicator values.

**Table 1 sensors-24-03873-t001:** Participants’ characteristics.

	Group 1 (*n* = 7)		Group 2 (*n* = 15)	Group 3 (*n* = 66)	
	Condition—Capsaicin		Condition—Thermal Stimulus	Condition—Chronic Pain	
	No Pain	With Pain	Thermal Pain	Centrally Acting Medication (*n* = 36)	Other Treatment (*n* = 30)
Average age (years)	32.40		31.70	41.17	52.10
Average of pain duration (months)	–		–	55.34	105.03
Average pain scores (Numerical scale/10)	0.00	4.00	Decrease in pain from 6.1 to 0.7	2.63	4.10

**Table 2 sensors-24-03873-t002:** Descriptive statistics of group 1 (*n* = 7).

ID	Sex: 1 = Male 2 = Female	Age (Years)	Pain Duration (Month)	Pain Scores (/10)		Piq_β_ (%)	
				No Pain Condition	With Pain Condition	No Pain Condition	With Pain Condition
1	1	35	–	0	5	7.13	12.35
2	2	45	–	0	5	8.28	11.23
3	1	33	–	0	4	7.16	13.00
4	1	30	–	0	2	8.15	12.54
5	1	28	–	0	5	8.24	13.56
6	1	24	–	0	5	7.13	14.12
7	1	26	–	0	2	9.00	10.15
**Mean (SD)**		**31.6 (7.0)**		**0 (0)**	**4 (1.4)**	**7.83 (0.74)**	**12.42 (1.4)**

**Table 3 sensors-24-03873-t003:** Descriptive statistics of group 3 (*n* = 66).

ID	Sex: 1 = Male 2 = Female	Age (Years)	Pain Duration (Month)	Pain Score (/10)	Piq_β_ (%)	Medication with Central Effect: 1 = Yes 0 = Other Medications
1	1	21	30	0.00	8.81	0
2	2	40	37	4.36	15.75	1
3	1	32	18	2.36	14.05	1
4	2	59	29	1.05	10.18	0
5	2	64	120	1.09	10.76	1
6	2	63	396	6.00	19.17	1
7	2	58	5	2.73	14.29	1
8	2	63	36	0.64	8.25	1
9	1	30	4	5.27	24.99	0
10	1	68	120	4.55	15.05	1
11	1	69	24	0.09	9.33	0
12	1	78	6	1.00	12.66	0
13	1	30	5	0.36	7.84	0
14	2	17	6	4.64	13.74	0
15	2	28	84	3.00	13.65	0
16	2	21	12	1.00	11.92	0
17	2	22	6	0.55	7.92	0
18	2	26	84	4.45	18.68	0
19	2	54	4	5.77	12.34	0
20	1	42	22	3.82	11.34	0
21	1	66	72	2.00	10.19	0
22	2	59	84	0.09	6.93	1
23	1	21	4	2.27	11.12	0
24	2	19	7	2.00	12.30	0
25	1	23	1	1.27	12.98	0
26	1	22	204	2.91	12.59	0
27	2	60	2	1.77	10.74	0
28	2	57	1	2.91	11.78	1
29	2	31	8	4.23	22.41	1
30	2	52	0.75	2.27	12.48	0
31	2	33	7	6.91	15.57	0
32	2	37	12	0.86	7.59	1
33	2	22	120	2.00	11.89	0
34	2	36	2	0.00	9.39	0
35	2	59	84	2.00	13.02	1
36	2	63	6	4.00	14.82	0
37	2	64	3	0.00	5.66	1
38	2	15	6	4.00	16.59	0
39	1	32	16	10.00	18.9	1
40	1	45	96	8.00	17.7	1
41	2	44	24	2.10	14.9	1
42	2	45	240	7.50	19.1	1
43	2	46	240	4.00	15.5	0
44	2	47	48	4.00	11.4	0
45	2	32	216	0.50	12.9	0
46	2	45	24	5.70	17.5	1
47	2	71	120	10.0	29.2	1
48	1	56	4	3.50	11.15	0
49	2	27	60	1.60	16.1	1
50	1	68	120	6.80	15.1	1
51	2	73	24	3.40	13.8	0
52	2	55	240	2.10	15.1	1
53	1	45	48	5.70	20.3	0
54	2	62	60	2.10	16.2	1
55	1	66	144	3.00	17.7	0
56	1	65	3	2.20	13.3	1
57	2	49	192	1.20	10.8	0
58	2	61	360	7.20	13.9	1
59	1	60	84	5.80	12.44	1
60	1	53	5	0.00	12.68	0
61	2	44	48	7.00	16.78	1
62	2	59	120	1.10	13.94	1
63	2	59	300	2.00	17.78	0
64	2	24	96	7.00	23.67	1
65	2	61	516	5.00	32.10	1
66	2	27	24	5.00	18.23	0
**Mean (SD)**		**46.1 (17.3)**	**77.9 (105.9)**	**3.30 (2.50)**	**14.3 (4.8)**	

**Table 4 sensors-24-03873-t004:** Comparative results. * Indicates statistical differences (*p* < 0.05).

Variables	Sub-Groups		*p* Value	Effect Size	Clinical Difference Δ (%)
	Medication with Centrally Acting (*n* = 36)	Medication without Centrally Acting (*n* = 30)			
Pain scores (/10)	2.63 (1.87) IC: 1.9–3.2	4.10 (2.91) IC: 3.0–5.1	0.016 *	0.61	55.8%
Piq_β_ (%)	13.23 (3.66) IC: 11.9–14.6	15.68 (5.80) IC: 13.5–17.8	0.041 *	0.51	18.5%

## Data Availability

The raw data supporting the reported results cannot be publicly shared due to confidentiality and their inclusion in an ongoing patent application, submitted in March 2024.

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
