# Peer review of "An Innovative EEG-Based Pain Identification and Quantification: A Pilot Study"

_sensors, 2024, doi:10.3390/s24123873_

Round 1
Reviewer 1 Report
Comments and Suggestions for Authors
- How does the indicator Piqβ evolve following a decrease in pain caused by the application of the cream?
- How is the criterion Piqβ=10% selected as a threshold for pain?
- volunteers in group 2 is applied pain stimulation at the beginning of the experiment. What will happen to the indicator "Piqβ" when the feeling of pain increases?
- For the third group of subjects, since many of them are suffering from continuous pain, especially those with chronic pain, if Refβ is considered as the first 60 seconds of EEG collection from the participant, it means that the pain-affected EEG has been counted into the normalized EEG data. I wonder what value of Piqβ will be obtained if the subject's pain suddenly stops and lasts for a few hours.
- What do the solid and hollow points in Figure 6 represent?
- There is an apostrophe symbol in Equation (1), what does it mean?
Reviewer 2 Report
Comments and Suggestions for Authors
I think the paper is well-written and this is a great work. But I think the introduction can offer a more detailed background.
Here are my comments.
Please divide the introduction into the background and objective for clarity.
Please reveal electrode locations.
The authors stated that “The second substep of filtering was to remove the 156 remaining artifacts due to eye blinks or eye movements, and the electromyography (EMG) 157 signal and all well-known noise, such as poor electrode contact quality, were identified 158 during the experiments”, but they did not explain how to remove.
For the selection of beta band, I think the authors should explain how the beta band is related to pain identification. As this plays a role throughout the paper, I think the authors should provide a profound background in the introduction too.
Usually, FIR filters are used for selecting frequency band in EEG analysis. Is there a specific reason to use an IIR filter?
Why did the authors adopt instantaneous amplitude? What did you expect?
Round 2
Reviewer 2 Report
Comments and Suggestions for Authors
Good job.